# COVID-19 and Frailty

**DOI:** 10.3390/vaccines11030606

**Published:** 2023-03-07

**Authors:** Tiziana Ciarambino, Pietro Crispino, Giovanni Minervini, Mauro Giordano

**Affiliations:** 1Internal Medicine Department, Hospital of Marcianise, ASL Caserta, 81037 Caserta, Italy; 2Internal Medicine Department, Hospital of Latina, ASL Latina, 04100 Latina, Italy; 3Internal Medicine Department, Hospital of Lagonegro, AOR San Carlo, 85042 Lagonegro, Italy; 4Department of Advanced Medical and Surgical Science, University of Campania, L. Vanvitelli, 81100 Naples, Italy

**Keywords:** SARS-CoV-2 infection, COVID-19, elderly, frailty, pandemic

## Abstract

Older age is a major risk factor for adverse outcomes of COVID-19, potentially due to immunosenescence and chronic low-grade inflammation, both characteristics of older adults which synergistically contribute to their vulnerability. Furthermore, older age is also associated with decreased kidney function and is consequently associated with an increased risk of cardiovascular disease. All of this in the course of COVID-19 infection can worsen and promote the progression of chronic kidney damage and all its sequelae. Frailty is a condition characterized by the decline in function of several homeostatic systems, leading to increased vulnerability to stressors and risk of adverse health outcomes. Thus, it is very likely that frailty, together with comorbidities, may have contributed to the high vulnerability to severe clinical manifestations and deaths from COVID-19 among older people. The combination of viral infection and chronic inflammation in the elderly could cause multiple unforeseen harmful consequences, affecting overall disability and mortality rates. In post-COVID-19 patients, inflammation has been implicated in sarcopenia progression, functional activity decline, and dementia. After the pandemic, it is imperative to shine a spotlight on these sequelae so that we can be prepared for the future outcomes of the ongoing pandemic. Here, we discuss the potential long-term consequences of SARS-CoV-2 infection and its possibility of causing permanent damage to the precarious balance existing in the frail elderly with multiple pathologies.

## 1. Introduction

Frailty is a condition characterized by the decline in function of several homeostatic systems leading to increased vulnerability to stressors, and risk of adverse health outcomes [1,2]. Thus, it is very likely that frailty, together with comorbidities, may have contributed to the high vulnerability to severe clinical manifestations and deaths from COVID-19 among older people. Frailty may have contributed to older persons’ vulnerability to a more severe clinical presentation. However, frailty has only been investigated for its association with overall mortality, hospital contagion, intensive care unit admission rates, and disease phenotypes, and finally for specific interventions related to frailty or its impact on COVID-19 treatments. The few observational studies retrieved have not yet been evaluated, or rather there is still limited evidence on the impact of COVID-19 in the determinism of complications related to frailty, which can then lead to an escalation of aggravating events up to death. It is now known, from data that have come in since the first moment of the pandemic in Wuhan, that the majority of COVID-19 cases have occurred in people aged 60 and over, and that death rates have increased exponentially with age, from 0.4% among those aged 40–49 to 3.6% among those aged 60–69, and 14.8% among those aged >80 [3]. Additional reports from Italy and the United Kingdom, which were among the hardest-hit countries in Europe, confirmed the high risk of death in the elderly, particularly those with pre-existing diseases such as cardiovascular and respiratory disease, obesity, diabetes, chronic kidney disease, gastrointestinal, skin, musculoskeletal, immune diseases, and cancer [4,5,6]. The conceptual framework of frailty describes a condition of increased vulnerability to stress factors due to the decline in the function of homeostatic mechanisms, and the consequent increase in the risk of adverse health outcomes [1,2]. Thus, it is very likely that frailty, together with comorbidities, could contribute to the high vulnerability and increased risk of death of older people infected with COVID-19. The definition of frailty has primarily a preventive meaning, as frailty assessment can help clinicians to identify those older people who need close monitoring of their functional status, and specific interventions aimed at reducing the risk of adverse outcomes. A growing body of evidence indicates that COVID-19 can manifest in atypical presentations, especially in the elderly. In Italy, 24% of patients with COVID-19 who died during the pandemic did not present with fever, 27% did not present with dyspnea, and 61% did not present with a cough at onset [7]. There is also the description of older people with COVID-19 who have a history of falls or delirium, further suggesting the need for early frailty assessment and careful monitoring of physical and cognitive status during a period of social distancing and self-isolation [8,9]. Sarcopenia, progressive functional deterioration, cognitive decline, and depression are elements not yet well understood in patients with COVID-19, for which there are a lack of primary and rehabilitative action plans capable of promoting an increase in life expectancy in patients’ post-infection [10]. Therefore, the aim of this systematic review is to summarize the evidence available in the literature on how COVID-19 infection and frailty enter into synergy with each other from an etiopathogenetic point of view, in an attempt to identify the mechanisms that promote the functional decline of the frail subject until death. The purpose of this review is to understand how acute disease manifestations are related to pre-existing health conditions, and how COVID-19 infection can lead to adverse outcomes during hospitalization or immediately after discharge. Understanding the mechanisms by which COVID-19 contributes to worsening frailty could lead to more informed decision-making for both patients and clinicians in terms of triage and eligibility for therapy. Furthermore, a proper assessment of frailty in patients with co-infecting COVID-19 would help to better distinguish those who will necessarily have to be hospitalized, from the others, for whom home care could be an option. This could lead to an individualized approach in the management of the current disease because some interventions could be not only less effective clinically, but perhaps more harmful.

## 2. Physical Decline in the Elderly and in Elderly with COVID-19

### 2.1. Malnutrition and Sarcopenia

In general, malnutrition and sarcopenia are the cause and result of functional and structural alterations and development of the organism, resulting from the imbalance between needs, income, and use of nutrients such as to lead to an excess of mortality and morbidity or an alteration in the quality of life [11]. These two conditions tend to occur physiologically over the years but are also common events determined by the set of acute, subacute, or chronic diseases, often characterized by an underlying chronic inflammatory state. This leads to a change in body composition and multiple organ dysfunctions [12]. The importance of inflammation as a factor capable of increasing the risk of malnutrition is supported by numerous scientific pieces of evidence, which show a significant production of numerous pro-inflammatory cytokines which elicit various physiological functions and can have metabolic depressive effects on the human body [11,12]. Tumor necrosis factor-α [TNF-α] participates in muscle proteolysis, and is able to modulate hepatic protein synthesis, glycogenolysis, and gluconeogenesis, and is able to influence glucose clearance by increasing its speed, production, and accumulation of lactate, and lipogenesis. Interleukin-1 [IL-1] and Interleukin-6 [IL-6] more specifically influence hepatic protido-synthesis, gluconeogenesis, and glucose clearance, facilitating the synthesis of fatty acids. Interferon-gamma determines, in particular, lipolysis [13,14]. The relationship between inflammation and sarcopenia has deep roots. In fact, as mentioned earlier, older adults are at greater risk of adverse outcomes due to conditions associated with aging. In general, compared to younger adults, older adults have a decreased ability of their immune system to cope with infections, which is mainly a result of impaired immune response to pathogens [14]. This impairment of the immune system, which is associated with aging, is called immunosenescence [14]. In addition to the age-related chronic inflammatory state and immunosenescence, a drastic change in microbiota biodiversity contributes to immune system weakness among the frail elderly. Previous studies had shown that the increase in plasma levels of the main pro-inflammatory cytokines was related to the biodiversity of the microbiota of elderly people, and where in some individuals there was a reduced diversity of the intestinal microbiota, this was associated with greater frailty [15,16,17]. It is therefore implicit that the degree of frailty, sarcopenia, immunosenescence, and changes in the intestinal microbiota are related to each other, and are affected by the progression of chronological age [18].

### 2.2. COVID-19 and Sarcopenia

As we have pointed out so far, there is a close relationship between nutrition, sarcopenia, and a chronic inflammatory state. As far as COVID-19 is concerned, observations indicate that both during the acute phase of the infection and immediately after the serological negativization of the infection, there is a weight decrease [19,20] to which dehydration and loss of muscle mass contribute most [21,22]. This phenomenon was more evident in elderly patients affected by COVID-19 [22,23]. In these patients, sarcopenia can develop acutely, and therefore within the first 4 weeks or over months, following mechanisms that are not yet fully understood [24,25]. The factors that would come into play in determining sarcopenia in these patients are many, and concern the state of health and the degree of fragility existing before the infection, the intensity of the inflammatory process linked to COVID-19, anorexia [due to the loss of appetite resulting from acute illness, anosmia, and ageusia], physical inactivity, and the type of gut microbiota composition [26]. Immunosenescence is the main determinant of pre-existing clinical conditions of COVID-19 infection and, as we have seen, it is characterized by a delayed and reduced activation of the innate immune response with an ineffective or uncoordinated adaptive immune response, making COVID-19 infection more severe [27]. Immunosenescence of the innate immune system is characterized by reduced cellular superoxide production and phagocytosis capacity. The reduced ratio of naïve to memory cells and the expansion of mature cell clones characterize the immunosenescence of the acquired immune system [28]. It would seem that mitochondrial dysfunction results from this, with defects in phagocytosis and antigenic processing related to the composition of the intestinal microbiota and the generation of reactive oxygen species [ROS] in the mitochondria, their vacuolization, and enlargement [29]. Age-related changes in mitochondria and the mitochondrial pool add to the susceptibility of developing sarcopenia. Muscle, as a tissue with a high energy turnover, has a large pool of mitochondria that would make muscle fibers more susceptible to ROS damage and autophagy in the event of COVID-19 infection [30,31]. Added to this is an increase in damaged mitochondrial DNA [mtDNA], with reduced production of new mitochondria and increased autophagy [26]. The cytokine storm, with IL-6 and TNF-α, is the event that correlates the degree of inflammation to sarcopenia, creating a high potential for multiorgan damage, including in skeletal and cardiac tissue muscle [32]. The elevated concentrations of c-reactive protein [CRP], IL-6, and TNF-α are the indices that most strongly correlate with sarcopenia and frailty, as it is the acute inflammation induced by COVID-19 which promotes mitochondrial damage and interferes with iron homeostasis, thus promoting the decrease in an energy substrate necessary for the functioning of the mitochondria [33,34,35]. A loss of mitochondrial function forces the organism to derive its energy from anaerobic metabolism, thus generating ROS and increasing cellular susceptibility to damage and cellular death [26]. During the acute phase of the COVID-19 disease, taste and smell disorders caused severe dietary restrictions leading to malnutrition [36,37]. This dysfunction was not only recorded during the acute phase of the infection but also persisted months after the negativization, negatively affecting the resumption of an adequate diet [38]. This problem in elderly people adds to the reduced strength of the chewing muscles and tongue, and poor salivation [39]. Difficulty swallowing often contributes to malnutrition and may be linked both to the reduced efficiency of the age-related mechanisms of swallowing and to sarcopenia induced by COVID-19, creating a vicious cycle of events [40]. On the other hand, infection increases calorie requirements, and in the absence of adequate dietary intake, this contributes to sarcopenia. Finally, the gastrointestinal and hepatic manifestations related to COVID-19 can further contribute to anorexia but also to the reduced assimilation of nutrients [41,42]. Sarcopenia is also related to bed rest secondary to COVID-19 symptoms, with loss of volume and muscle mass and functional impairment [43,44]. Similarly, hospitalization times are also related to the loss of lean mass and therefore muscle tone, especially if linked to other chronic conditions or concomitant traumas [45]. Finally, we underlined the role of gut microbiota in promoting nutrition and sarcopenia in patients with COVID-19 [46], modulating the inflammatory process, determining its severity, and influencing the processes of immunosenescence. We currently know only partially the impact of the interaction between COVID-19-related inflammation and the gut microbiota. It has been observed that a microbiota characterized by anaerobic bacteria is associated with increased severity of the sequelae of infection, while other species with antiphlogistic activity were correlated with a more indolent course of the disease [46]. In particular, in the frail elderly, the susceptibility to infection depended on the interaction between the ability of the bacterial species to sensitize the immune system associated with the mucosa, with the biodiversity of the species that make up the microbiota. Hence, reduced microbiota diversity accompanied by immunosenescence and inflammatory aging played an important role in promoting severe forms of COVID-19 infection [46,47].

## 3. Functional Decline in the Elderly and in Elderly with COVID-19

### 3.1. Elderly Functional Deterioration

Patients with progressive functional deterioration are individuals who often lack therapeutic resources that can be proposed to extend life expectancy and quality, placing the burden of assistance on the caregiver. For the elderly person with growing care needs and illness, this condition affects their family and social life too. The sick person can no longer be alone, not even for a short time, and in this phase, the exact understanding becomes problematic as it impacts the needs of the families and health personnel. Failure to recognize these needs quickly leads to the aggravation of the main disease and co-morbidities, thus accentuating frailty. The exhaustion of functional capacities is the first indicator of frailty. Global fatigue is characterized by a general subjective lack of physical or mental energy, it can limit physical activity, and lead to sedentary behavior, which further deteriorates physical functioning and, over time, increases the risk of functional limitation and mortality [48]. As a risk factor distal in the causal chain, perception of exhaustion or fatigue has been hypothesized to be one of the earliest markers identifying people at risk for frailty [49]. The exhaustibility of physical capabilities is the trigger for behavioral changes, which predispose people to a vicious cycle that ultimately leads to frailty. Chronic inflammation has been suggested as the link between sarcopenia and progressive functional deterioration. Especially in older subjects, the high level of inflammatory factors can influence acute changes in body size, in particular, the amount, structure, and function of skeletal muscles which would summarily amount to sarcopenia and consequently to increased exhaustibility, reduced tolerance to physical effort, greater dependence, and therefore more frailty. The continuous activation of the immune system, tissue repair, and attempts to maintain correct body homeostasis are all inflammatory triggers that determine the functional decline of the frail person. The state of chronic inflammation negatively affects individual performance through increased oxidative metabolism and increased cytokine cascade [50].

### 3.2. COVID-19 and Functional Deterioration

There is no doubt that the COVID-19 infection has a significant impact on the functional decline of frail subjects, precisely by virtue of the chronic systemic and subclinical inflammation and the impairment of the acquired immune system [51]. COVID-19 involves an infection mediated by a hyperinflammatory response associated with cytokine release syndrome [52,53]. The control of the vicious cycle that transforms a normal cytokine response into a cytokine storm has been one of the turning points in the control of the infection and its sequelae, with a progressive decrease in the associated mortality [52,53]. The hyperinflammation promoted by COVID-19 is also associated with pyroptosis, an inflammatory form of apoptosis triggered by viral replication. This programmed cell death exacerbates the inflammatory response by releasing IL-1B from dying cells [54]. This overproduction of cytokines, in addition to acutely causing the most severe phenotype of the disease through various mechanisms, pulmonary lesions, and microthrombi, also leads to the functional exhaustion of immune cells over time, causing the permanence of overactivation of the immune system [55,56]. The functional decline related to the COVID-19 infection is associated with other sequelae that have the same point of origin as sarcopenia; the post-infection cardiovascular, pulmonary, and psychological repercussions can delay or nullify the recovery phase after the acute event [57,58]. One study showed that 87.4% of patients had at least one disorder attributable to each of these sequelae that persisted into the post-COVID-19 phase. Most of these symptoms included dyspnea or fatigue, while there were also a number of patients presenting with arthralgias or pain in other parts of the body [59]. One study indicated that in patients with COVID-19, postinfectious physical function and fitness may be impaired for up to two years after illness [60]. These factors may purely mechanistically adversely affect physical performance, and thus aggravate all other sequelae including sarcopenia and emotional disorders [61,62]. In fact, a deflection of mood also has negative consequences on physical activity [63]. The disappearance of dyspnea is a delayed event in many patients and results in less physical activity [64].

## 4. Cognitive Decline in the Elderly and in Elderly with COVID-19

### 4.1. Cognitive Decline and Depression

Dementia is considered a fatal disease, even if it is not among the main causes of death [65]. This is because dementia is not reported in the death certificates or medical records, and instead codes for a terminal disease, e.g., pneumonia, are recorded, neglecting to specify dementia as the underlying cause. It is therefore estimated that deaths from dementia are at least five to six times higher than that reported among people over 75 years of age, and therefore may often be the cause of death in these patients. The treatment of cognitive deficits in the initial and moderate phases of the disease has been benefiting for the past 25 years from acetyl-cholinesterase inhibitors (donepezil, rivastigmine, galantamine), which increase the levels of acetylcholine in the synaptic cleft. A therapeutic response is obtained in 20–40% of patients, with the effect of slowing down the progression of the disease and the consequent loss of independence. In the more advanced stages of moderate to severe disease with Mini-Mental State Examination (MMSE) between 10 and 19/30, memantine, an antagonist at uncompetitive glutamate receptors in the brain, is used, which reduces the pathological input of calcium ions in resting conditions and allows their entry following synaptic activation [66]. The terminal phase of the disease (MMSE < 10) does not require the use of drugs capable of affecting the natural history of the disease. Instead, the treatment is limited to relieving the complications related to the neurodegenerative disorder such as cachexia, decubitus, and infectious complications which subsequently cause the death [67]. Depression is a symptom of the onset of dementia, assuming that there are etiopathogenetic analogies between depression and degenerative dementia with coexisting cerebrovascular disease [68]. However, it can sometimes occur due to the awareness of illness in the initial stages. In the frail elderly, it is more complex to distinguish the onset of this condition since it occurs more frequently with non-specific symptoms such as asthenia, insomnia, and hypochondria [69]. The best outcomes are obtained in the initial stages when the patient is aware of their symptoms and their condition; while, in the terminal stages, instead, there is no rationale in treating the disorders in the absence of a patient’s real awareness of their condition [69]. Behavior disorders such as irritability, aggression, motor activity purposelessness, delusions, and hallucinations are frequent in the moderate-severe stage of dementia, and their occurrence in elderly patients is a sign of increased frailty; therefore, a specific drug treatment may be appropriate for use from symptom onset to the terminal stages of life. These disorders are often the causes of improper hospitalization, as the environmental changes that hospitalization entails lead to the need for patient restraint, either physical or pharmacological, making it difficult to manage their main vital functions. Prolonged sedation, progressive bed rest, dehydration, and malnutrition often distract from other pathologies such as infections, sepsis, thromboembolic complications, and irreversible electrolyte imbalances, which with a cascade mechanism can often lead to exitus [70]. Before starting any pharmacological treatment aimed at these disorders, it is necessary to identify and remove the possible causes and triggers of behavioral disorders. Especially in terminally ill patients, there may be physical causes, such as pain, infections, urinary retention, and constipation, which the patient may express with behavioral disturbances, but which instead require an etiological and non-symptomatic treatment [71,72,73]. In advanced-stage dementia, particular attention must be paid to the effects of the environment when staying in new spaces or spaces not recognized by the subject. These effects may include an excess of strong light or sound stimuli, or, on the contrary, the deprivation of light, as well as adverse attitudes of the caregiver or modification of the caregiver habits, the presence of strangers in the hospital, and the absence of habitual contact with family members. These are all factors which may be the triggers of behavioral disorders [71,72,73].

### 4.2. COVID-19 and Cognitive Decline

It is known that aging is the most significant risk factor for the development of neurodegenerative diseases. Inflammation in the central nervous system (CNS), i.e., neuroinflammation, plays a fundamental role in the severity of the pathogenesis of these pathologies [73]. Neuroinflammation is primarily regulated by glial cells, such as microglia and astrocytes. The first cells are considered to be resident macrophages of the brain, thus representing the first line of immune defense in the central nervous system. In addition, they have a role in the elimination of toxic substances, damaged cells, and pathogens, thus regulating both the pro-inflammatory and anti-inflammatory responses [74]. During pathogenesis, microglia become activated due to cell damage and the presence of protein aggregates in their environment, triggering the production of chemokines and cytokines [75]. The resulting oxidative stress amplifies the damage to cellular components and further activates neighboring glial cells, thereby resulting in a chronic inflammatory state [76]. Although the data regarding the relationship between chronic inflammation and degenerative neurological damage are very clear, the role of chemokines in COVID-19 infection has not yet been clarified. The virus uses lymphatic circulation to colonize the nervous system through infection of lymphatic endothelial cells [77,78]. The presence of the virus has been confirmed in neuronal and capillary cells in the frontal lobe of COVID-19 patients, associated with the worsening of neurological symptoms [79]. Aging triggers debilitating conditions, such as low-grade systemic inflammation and neurodegeneration. Such conditions can be triggered or aggravated by viral infections. It has also been proposed that COVID-19 infection may disrupt cellular homeostasis, ultimately leading to protein misfolding and, in this way, increasing the propensity for the future development of neurodegenerative diseases [80]. The loss of the ability to correctly activate stress response mechanisms in the elderly is one of the mechanisms associated with neurodegenerative disease, and is one of the factors linking the fragility of these subjects to COVID-19 infection [80,81]. A particular category of patients that we need to focus on when talking about dementia in the pandemic period is that of patients with Down syndrome (DS). The latter experience a progressive cognitive deterioration in the first years of life and with the increase in their life expectancy due to advances in the medical field, there is a high prevalence of dementia cases at age 65 [82]. Furthermore, the COVID-19 pandemic in this population has been even more violent, not only due to the restrictions linked to the prolonged lockdown which has prevented their relationship with the outside world and with activities in society, but from a clinical point of view it is also linked with the presence of a deficient immune system, and therefore susceptibility, in the case of infection, to possible exacerbations of functional and cognitive deterioration [83]. Villani et al. [82] have demonstrated that the pandemic and the discomfort associated with it due to the related social isolation measures have led to a worsening of depressive symptoms and have led to a functional decline in a selected population of adults with DS. The increase in depressive symptoms was only minimally related to an increase in the incidence of aggressive traits.

## 5. Frailty and COVID-19 Vaccination

Chronic inflammation and immunosenescence also seem to be associated with an altered and highly variable immunological response to vaccination [84,85]. In particular, a poor response to vaccines has been noted in frail people [86,87]. It is therefore possible that despite vaccination or due to their potential poor response to vaccines, frail elderly people are still at risk of contracting the infection or having a recurrence of the infection, perhaps associated with different variants of COVID-19 [88]. Salvagno et al. [89] have shown that in most cases of poor immunization and vaccine responsiveness, mainly the frailest categories of patients are affected, i.e., those who have a higher risk of complications of the disease [19]. In the study, it was observed that total anti-SARS-CoV-2 receptor-binding domain (RBD) antibodies were inversely associated with age after the first two vaccine doses, with male sex after the second vaccine dose, and in seronegative subjects at baseline. Hence, the immune response to the post-mRNA COVID-19 vaccine in seronegative subjects or those already seropositive at baseline would appear to be associated with age and gender in seronegative subjects, and additionally by the baseline anti-SARS-CoV-2 antibody level in seropositive patients. The lower total anti-SARS-CoV-2 RBD antibody response found in males and older baseline seronegative subjects would suggest that vaccines confer less protection against infection, and a higher risk of developing more aggressive forms of COVID-19. Another study [90] highlights how the antibody titer can be kept high using all three doses of the vaccine, emphasizing that the same titer is higher in patients already immunized with the first dose. In addition, according to this study, the frailest patients show an overall reduced response to the anti-SARS-CoV-2 vaccination compared to healthy individuals, with a significantly greater decline in antibody titers within the first 6 months.

## 6. Conclusions

Inflammation has been implicated in the progression of sarcopenia, frailty, and dementia (Table 1). Especially in older subjects, the high level of inflammatory factors, as observed in COVID-19, may influence acute changes in body size, especially the amount, structure, and function of skeletal muscles which would summarily amount to sarcopenia. Older people and people with comorbidities are frail individuals and are more prone to showing severe symptoms of COVID-19 due to cellular senescence in affected tissues and synergistic activation in the immune system by senescence processes and cytokine reactions to the virus. In the frail, COVID-19 exploits the weakness of the organism and the deficiency of the immune system by amplifying inflammation associated with aging and chronic disease, and also increases cytokine production in the nervous system, thus increasing the risk of developing neurodegenerative diseases or worsening them if already present. Normal mitochondrial function and responses to stress are just some of the cellular pathways COVID-19 uses that are affected by the infection. This behavior from the virus suggests that aging is accelerated in these patients and could be a potential long-term complication of COVID-19 infections, associated with a greater risk of frailty and therefore greater mortality. Finally, as regarding the efficacy of vaccines, it has been observed that advanced age, frailty degree, and male gender significantly influence the immunogenicity induced by the vaccine. In particular, males and those of advanced age appear to exhibit a lower efficacy humoral response.

## Figures and Tables

**Table 1 vaccines-11-00606-t001:** Take-home messages about COVID-19 infection and frailty.

Take-Home Messages
Inflammation is implicated in the progression of sarcopenia, frailty, and dementia in elderly subjects with COVID-19.
Older people with comorbidities are frail individuals and are more prone to showing severe symptoms of COVID-19 due to synergistic activation of the immune system and virus-induced senescence processes.
In the frail, COVID-19, by stimulating the immune system and amplifying inflammation associated with aging and chronic diseases, increases the risk of worsening already existing neurodegenerative diseases.
The behavior of the COVID-19 virus suggests an acceleration of aging processes and potentially an aggravation of comorbidities, greater risk of frailty, and therefore greater mortality.
The efficacy of vaccines decreases significantly with advanced age, frailty degree, and male gender. These variables appear to be associated with a lower efficacy humoral response and more severe prognosis in case of infection.

## Data Availability

Not applicable.

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
