# Peer review of "COVID-19 and Frailty"

_vaccines, 2023, doi:10.3390/vaccines11030606_

Round 1

Reviewer 1 Report

I read with very interest the manuscript entitled “COVID-19 and frailty”, on an argument of great interest, like as the impact of COVID-19 in the elderly. The Authors discuss about the potential long-term consequences of SARS-CoV-2 infection and its possibility of causing permanent damage to the precarious balance existing in the frail elderly with multiple pathologies.

The paper is well written and interesting to read, however I see the following major issues that should be resolved before publishing this paper: 

·      I suggest of considering removing the paragraphs Methods and Results as it is a review.

·      There are paragraphs I had trouble following as they are too long to read, I suggest putting together the paragraph on aging condition and COVID-19, for example “malnutrition and sarcopenia” and “COVID-19 and sarcopenia”.

·      I suggest discussing more about “frailty and COVID-19 vaccination”. Recently, some Authors evaluated the total levels of anti-SARS-CoV-2 RBD antibodies after vaccine administration suggesting that the humoral immune response to vaccine is age‐dependent and gender‐dependent with elderly having significantly lower levels of antibodies than young subjects. Please, briefly discuss about them (PMID: 34064509, PMID: 35606426; PMID: 34206567)

·      I recommend avoiding the use of the abbreviation the first time it was used in the text. After you define an abbreviation (regardless of whether it is in parentheses), use only the abbreviation. Do not alternate between spelling out the term and abbreviating it. For example, line 99 (IL-6) and line 144. Please check the whole manuscript.

·      Please carefully manuscript spell check to eliminate grammatical error, for examplelines 35 and 40.

·      Moreover, I suggest a graphical abstract to quickly gain an overview on primary concepts of the study.

Author Response

Naples, 1 March 2023

Dear Professor Doctor

Vaccines Editorial Office

Please, find enclosed the revised version of the manuscript entitled: “COVID-19 AND FRAILTY” We thank the Editor and the Reviewers for their comments and we hope that the following changes will now make the manuscript suitable for publication on the Vaccines. Please see the following list of the changes made in manuscript in yellow.

REVIEWER 1

  • In accordance with the reviewer's suggestions, the Methods and Results paragraphs have been removed.
  • In accordance with the reviewer's suggestions, the paragraph on the condition of aging and COVID-19, has been suffixed to "malnutrition and sarcopenia" and "COVID-19 and sarcopenia".
  • In accordance with the reviewer's suggestions, the topic of "COVID-19 frailty and vaccination" was most discussed. Including recent articles in which the authors assessed total anti-SARS-CoV-2 RBD antibody levels after vaccine administration, suggesting that the humoral immune response to the vaccine is age- and sex-dependent, with the elderly who have significantly lower antibody levels than young people. (PMID: 34064509, PMID: 35606426; PMID: 34206567)
  • In accordance with the reviewer's suggestions, all abbreviations have been revised, especially in line 99 (IL-6) and in line 144.
  • In accordance with the reviewer's suggestions, the spelling of the manuscript was checked especially in lines 35 and 40.
  • In accordance with the reviewer's suggestions, a table on the primary concepts of the study was made.

Best regards,

Tiziana Ciarambino

Reviewer 2 Report

Dear Authors, the presented study is something that we are needing and it is well organized and easy to read. Overall, I think that it be both useful for researchers/clinicians and for the visibility of the Journal.

Actually I do not really have to ask for you to change.

I only leave you with a suggestion of improvement: when you write of cognitive decline and depression and COVID-19 and cognitive decline, I think that you could include the peculiar category of the adults with DS.

They are a sort of geriatric patient, and COVID-19 related lockdown had a huge effect on their mood, functioning and cognitive decline, as shown in the paper  

Impact of COVID-19-Related Lockdown on Psychosocial, Cognitive, and Functional Well-Being in Adults With Down Syndrome

 37 subjects of the study sample, and a two time point evaluation (one pre- and one post-lockdown) in 9 subjects. Two mixed linear regression models - one before and one after the lockdown - have been fitted for each scale in order to investigate the change in the time-dependent variation of the scores. In the pre-lockdown period, significant worsening over time (i.e., per year) was found for the Depression Rating Scale score (β = 0.55; 95% CI 0.34; 0.76). In the post-lockdown period, a significant worsening in social withdrawal (β = 3.05, 95% CI 0.39; 5.70), instrumental activities of daily living (β = 1.13, 95% CI 0.08; 2.18) and depression rating (β = 1.65, 95% CI 0.33; 2.97) scales scores was observed, as was a significant improvement in aggressive behavior (β = -1.40, 95% CI -2.69; -0.10). Despite the undoubtful importance of the lockdown in order to reduce the spreading of the CoVID-19 pandemic, the related social isolation measures suggest an exacerbation of depressive symptoms and a worsening in functional status in a sample of adults with DS. At the opposite, aggressive behavior was reduced after the lockdown period. 

Adults with DS are now prevalent and their lifespan is increasing. I really think that they would deserve a bit of discussion in your paper

Author Response

Naples, 1 March 2023

Dear Professor Doctor

Vaccines Editorial Office

Please, find enclosed the revised version of the manuscript entitled: “COVID-19 AND FRAILTY” We thank the Editor and the Reviewers for their comments and we hope that the following changes will now make the manuscript suitable for publication on the Vaccines. Please see the following list of the changes made in manuscript in yellow.

REVIEWER 2

  • In accordance with the reviewer's suggestions in the section "cognitive decline and depression and COVID-19 and cognitive decline" a comment on patients with Down syndrome and a comment on the article was included: “Impact of COVID-19-related lockdown on psychosocial, cognitive, and functional well-being in adults with Down syndrome”

Best regards

Tiziana Ciarambino

Reviewer 3 Report

This mainly is mainly in the format of a review and the content is well suited for the format.

Abstract: Maybe also mention that Older age is also associated with reduced kidney function. (Reduced kidney function is associated with increased CVD risk and will reduce the clearance of low molecular weight proteins (e.g. many cytokines). Acute Kidney injury is also a major complication of COVID infections which may progress into chronic kidney disease.)

Line 35. I do not understand -19 in that sentence. Typo error?

Line 45-46: respiratory disease, obesity, diabetes, chronic kidney, cancer, gastrointestinal, skin, musculoskeletal and immune diseases. Disease after respiratory but not after chronic kidney. Cancer normally do not have disease after it. To me it seems a bit unsorted. I am a bit uncertain what the last diseases refer to. Please review the use of disease in the sentence

Many elderly in my country suffer from malnutrition and sarcopenia which probably makes them more vulnerable. Maybe strengthen the wording on the association between age and malnutrition/sarcopenia.

Line 78 Methods. Often the number of publications selected is usually mentioned for Pubmed searches.

Line 120: ). to which Typo error?

Line 146: tissue muscle What is included in this? skeletal muscle, cardiac muscle, and/or smooth muscle.  

Line 169: ). modulating Typo error?

Line 286: use of [] instead of () around reference.

Author Response

Naples, 1 March 2023

Dear Professor Doctor

Vaccines Editorial Office

Please, find enclosed the revised version of the manuscript entitled: “COVID-19 AND FRAILTY” We thank the Editor and the Reviewers for their comments and we hope that the following changes will now make the manuscript suitable for publication on the Vaccines. Please see the following list of the changes made in manuscript in yellow.

REVIEWER 3

  • In accordance with the suggestions of the reviewer it has been mentioned that “Advanced age is also associated with reduced kidney function. (Reduced kidney function is associated with an increased risk of CVD and will reduce the clearance of low molecular weight proteins (e.g., many cytokines). Damage also acute kidney injury is a major complication of COVID infections that can progress to chronic kidney disease.)"
  • In accordance with the reviser's suggestions the error in line 35 has been corrected.
  • In accordance with the reviewer's suggestions, the contents of Lines 45-46 have been better clarified.
  • In accordance with the reviewer's suggestions, the role of age on the malnutrition/sarcopenia association was strengthened.
  • Following the suggestion of the reviewer and based on the suggestion of the first reviewer the section on methods has been removed.
  • According to the suggestions of the reviewer the error of Line 120: ) has been corrected.
  • According to the reviewer's suggestions, the text on Line 146 has been better clarified
  • According to the suggestions of the reviewer the error Line 169: ). it was corrected
  • According to the suggestions of the reviewer the error Line 286 has been corrected.

Best regards,

Tiziana Ciarambino